# Disparities in physician compensation for breast and prostate cancer operations

**Noah Khosrowzadeh** [ID], **Aren Singh Saini** [ID]*, **Kayla Samimi, Kyle Chambers, Matthew Gompels, Cristiane Takita, Jessica Meshman, Cyrus Washington**

University of Miami Hospitals and Clinics: Sylvester Comprehensive Cancer Center, Miami, Florida, United States of America

* arensaini@med.miami.edu

## Abstract

The objective of this study was to determine whether the Center for Medicare and Medicaid services (CMS) pays more relative value units (RVUs) for prostatectomies than mastectomies across different treatment modalities. The RVU model is public information, subject to review every 5 years at minimum and has implications directly or indirectly to every medical center in the United States. These findings shed light on how the CMS values the two most prominent cancers among men and women: breast cancer and prostate cancer. An economic evaluation was conducted to appraise physician compensation for treating breast cancer vs prostate cancer. Work RVUs, malpractice RVUs, and practice expense RVUs were collected from the CMS' 2023 Physician Fee Scheduler. The total operative times, including time spent in pre-op consultations and follow ups, used by the RVU update committee to create these values were collected from the CMS' 2023 Final Rule Physician Work Times. Surgical oncologists treating breast cancer were paid an average of 15.45 RVUs per mastectomy and 2.74 per hour. The average work RVUs paid for a prostatectomy was 24.64 and 3.06 per hour. The average work RVUs paid for an axillary lymph node removal was 12.29 and 2.94 per hour. The average work RVUs paid for a pelvic lymphadenectomy was 15.87 and 3.04 per hour. The data assimilated in this study acts to illustrate the discrepancy between work RVUs for breast cancer and prostate cancer operations.

## 1. Introduction

In a vast array of different methods used to quantify healthcare processes, the Relative Value Unit (RVUs) has become the federal precedent set by the Centers for Medicare and Medicaid Services (CMS) to assign value to different procedures and processes. The RVU conversion factor in 2023 was $33.06 per unit [1]. Physician work RVUs, malpractice RVUs (MP RVUs) and practice expense RVUs (PE RVUs) were first implemented in 1992 as a system for assigning payment to physicians

**Data availability statement:** All relevant data are within the article and its supporting information files.

**Funding:** The author(s) received no specific funding for this work.

**Competing interests:** The authors have declared that no competing interests exist.

while providing administrative budgeting based on clinical and liability expenditures. These values are subject to reevaluation every 5 years at minimum or as new services become available. The CMS updates its RVU model by consulting the American Medical Association's Specialty Society Relative Value Scale Update Committee (RUC), which controversially sets higher values in certain fields of medicine than others. This article will objectively analyze the RVUs provided by the CMS to surgical oncologists treating breast cancer and compare them to the RVUs compensated to urologists treating prostate cancer, across multiple modalities.

## 1.1 Background

Breast cancer is the most diagnosed non-skin cancer in the world, comprising 12.5% of all malignancies across both genders in 2020 [2]. This number does not reflect the massively disproportionate incidence between males and females, with only 1% of identified cases being in men. In 2022, it is estimated that 287,850 new cases of invasive breast cancer and 51,400 new cases of ductal carcinoma in situ (DCIS) were identified among United States women, killing 43,250 women during that time [3]. The median age of diagnosis across women is 62 years old, and it is estimated that 1 in 8 women will be diagnosed with breast cancer in their lifetime with 1 in 39 women dying from breast cancer in the United States.

Treatment modalities for breast cancer are mostly radiotherapy, breast-conserving surgery, mastectomy, hormonal therapy and/or chemotherapy. Preferred treatment plans and adjuvant therapies widely vary based on staging, molecular subtypes, racial and socioeconomic factors. Overall, surgical oncologists are a pillar of care for the majority of stage I-III breast cancers (Fig 1) [3]. Partial mastectomies, also known as lumpectomies, aim to remove the abnormal tissue with minimal margins of surrounding healthy tissue. In a simple mastectomy, surgeons may remove the entire breast, areola, and nipple. In modified radical mastectomies, the entire breast will be removed along with most of the axillary lymph nodes. Surgical oncologists also conduct level 1 and level 2 axillary node dissections for further regional control of invasive breast cancer.

Prostate cancer is the leading diagnosed non-skin malignancy in men, with a large increase in cases in recent years with highly sensitive screening techniques such as the prostate specific antigen (PSA) test continuing to gain popularity. In 2022, it is estimated that 268,490 new cases of prostate cancer were identified in the United States, leading to 34,500 fatalities in that span [4]. Prostate cancer exclusively affects males, with the median age of diagnosis being 66. Much like breast cancer, prostate cancer affects 1 in 8 men throughout their lifetime, with about 1 in every 41 men dying from prostate cancer in the United States.

Treatment modalities for prostate cancer are primarily radiotherapy, robotic or open radical prostatectomy, hormonal therapy and/or chemotherapy. Preferred treatment plans are based on multiple factors, with staging and risk grouping being major indicators of which modality is selected. Again, a central pillar of combating localized prostate cancer is surgical. With newer surgical techniques, DaVinci robot-assisted radical prostatectomies have gained popularity over the traditional open radical

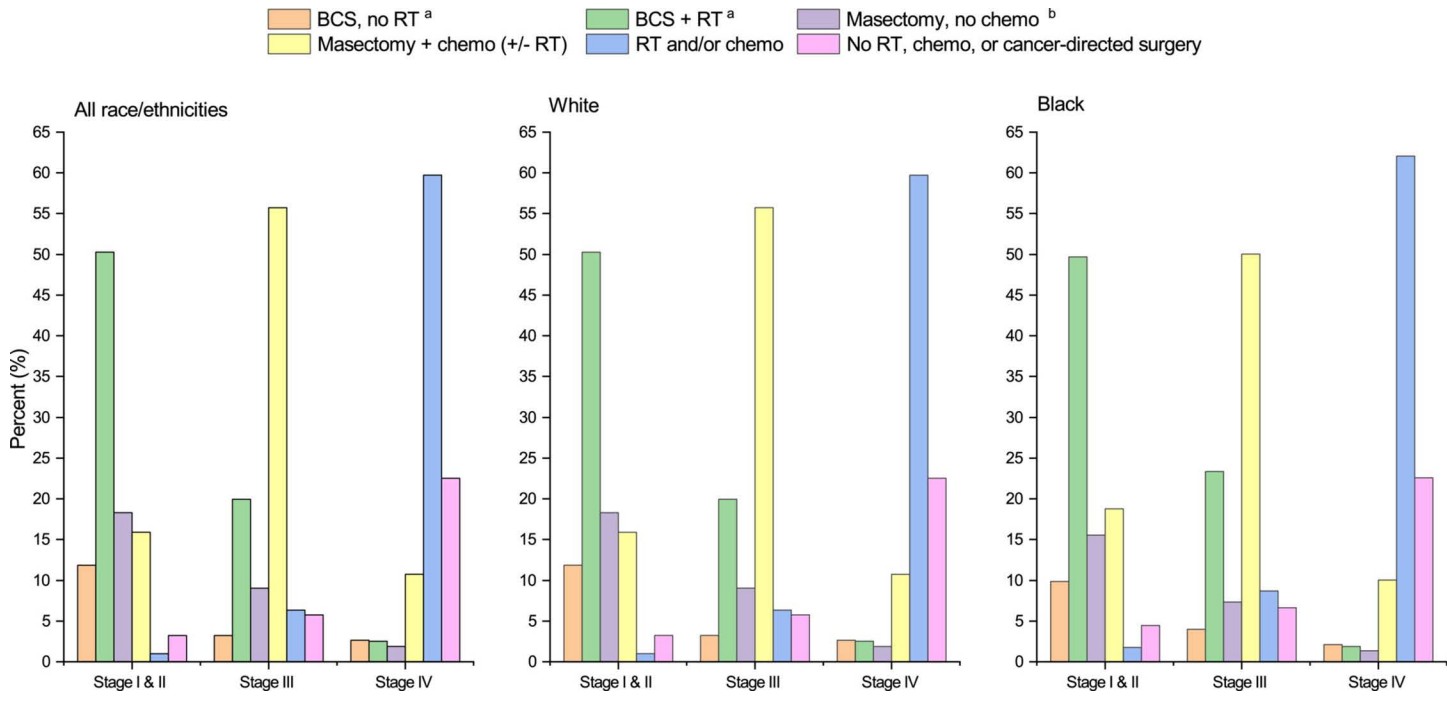

**Fig 1. 2018 Treatment Modalities for Breast Cancer [3].**

prostatectomy as laparoscopic operations are thought to decrease blood loss, have shorter hospital stays, and provide better postoperative urinary and sexual health outcomes [5]. Much like surgical oncologists in breast cancer, urologists often will perform pelvic lymphadenectomies to stage and control regional spread of prostate cancer.

## 2. Methods

Work RVUs, practice expense RVUs, and malpractice RVUs were obtained from the CMS 2023 physician fee schedule, using the HCPCS codes to identify the services from their database [6]. Estimated pre-evaluation times, intra-procedure times, immediate post service times, and total procedural times were gathered from the CY 2023 Final Rule Physician Work Time database provided by CMS [7].

The HCPCS codes used for mastectomy variations were 19301 for partial mastectomy, 19302 for partial mastectomy with lymph node removal, 19303 for a simple complete mastectomy, 19305 for a radical mastectomy, 19306 for an urban type radical mastectomy, and 19307 for a modified radical mastectomy. The HCPCS codes used for armpit lymph nodes were 38740 and 38745 for axillary lymph node dissections.

The HCPCS codes used for prostatectomy variations were 55810 and 55812 for a radical perineal prostatectomy, 55840 and 55845 for a radical retropubic prostatectomy, and 55866 for a laparoscopic retropubic radical prostatectomy. The HCPCS codes used for pelvic lymph nodes were 38571–38573 for laparoscopic lymphadenectomies.

Work RVUs were compared between mastectomy and prostatectomy procedures, and axillary node procedures were compared to those of pelvic lymph nodes. To further assess any true disparities in values assigned to the operations, a second analysis of work RVUs divided by total operation times was compared between their surgical counterparts (S1 Appendix).

## 3. Results

When examining the total work RVUs provided by the 2023 CMS physician fee schedule, there was an apparent difference between how the Medicare/Medicaid federal systems value gross physician compensation between mastectomies and prostatectomies (Fig 2). The average work RVUs paid for a mastectomy was 15.45, and the average work RVUs paid for a prostatectomy was 24.64. This indicates an average of 59% more work RVUs are compensated in prostatectomies than mastectomies per procedure. To further identify a potential disparity between the value CMS places on mastectomies as opposed to prostatectomies, the work RVUs were also compared as a factor of payment per hour of total procedure time (Fig 3). This analysis showed an average of 2.74 wRVUs per hour of total procedure time for mastectomies and an average of 3.06 wRVUs per hour of total procedure time for prostatectomies. On average, the CMS pays physicians 12% more for prostatectomies than mastectomies per hour.

When examining the total work RVUs provided by the 2023 CMS physician fee schedule, there was again a difference in the average work RVUs compensated for procedures involving the axillary lymph nodes in breast cancer and pelvic

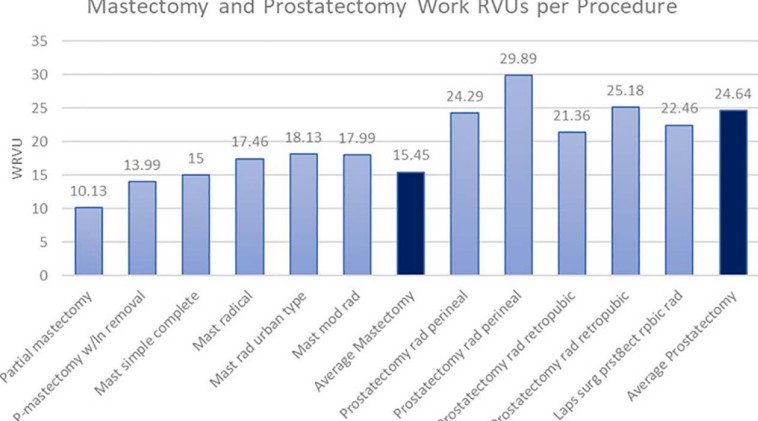

**Fig 2. Total work RVUs for mastectomy and prostatectomy procedures assembled from data obtained via CMS ( S1 Appendix).**

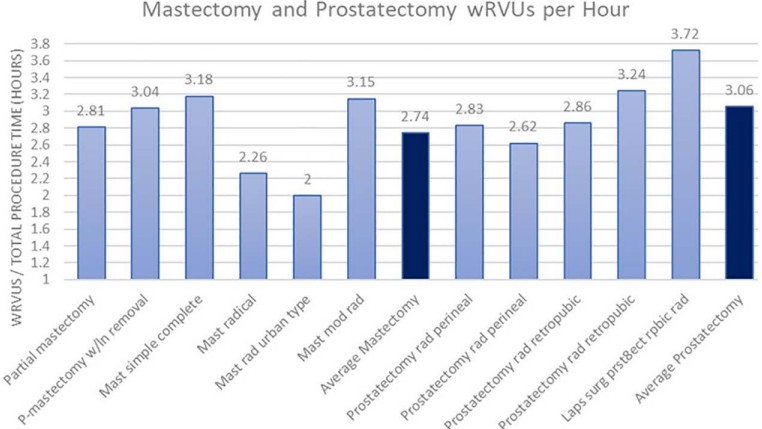

**Fig 3. Total work RVUs for mastectomy and prostatectomy procedures per hour of total procedure time assembled from data obtained via CMS ( S1 Appendix).**

lymph nodes in prostate cancer (Fig 4). The average work RVUs paid for an axillary lymph node removal was 12.29, and the average work RVUs paid for a pelvic lymphadenectomy was 15.87, indicating the CMS pays urologists 29% more per lymph node procedure than breast surgeons. When looking at the more accurate portrayal of payment discrepancy in the form of work RVUs per hour of total operative time (Fig 5), it was found that pelvic lymph node removals pay 3.04 wRVUs per hour and axillary lymph node removals pay 2.94 wRVUs per hour. This corresponds to a 3.4% higher payment to urologists than breast surgeons per hour for lymph node procedures.

In terms of the financial implications for medical institutions, the average practice expense RVUs for mastectomies stand at 11.085, lower than the corresponding 11.46 practice expense RVUs for prostatectomies. Similarly, the average practice expense RVUs for axillary lymph node removals was 8.43, compared to the 8.85 for pelvic lymph node removals.

Turning attention to malpractice coverage costs, mastectomies register an average of 3.81 malpractice RVUs, in contrast to the 2.94 for prostatectomies. Additionally, axillary lymph node removals demonstrate an average of 2.99 malpractice RVUs, compared to the 2.45 malpractice RVUs for pelvic lymphadenectomy.

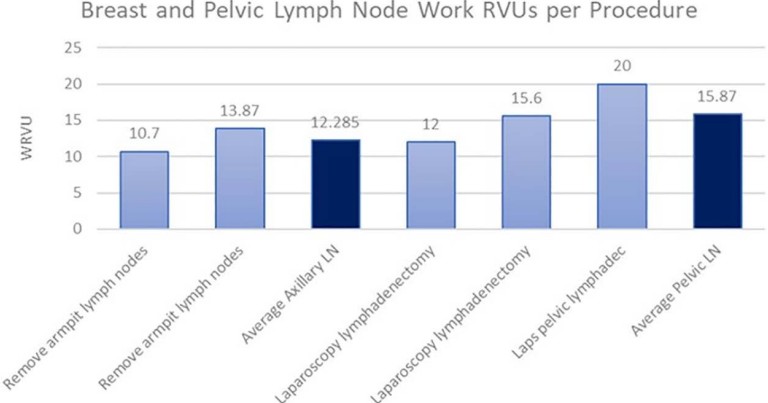

**Fig 4. Work RVUs for breast and pelvic LN procedures assembled from data via CMS ( S1 Appendix).**

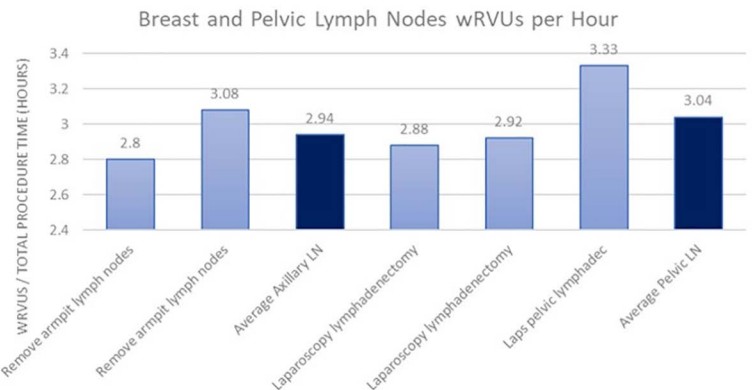

**Fig 5. Work RVUs for breast and pelvic LN procedures per hour total operative time ( S1 Appendix).**

## 4. Discussion

The findings from this comparative analysis of CMS physician compensation for breast and prostate cancer operations reveal significant disparities in how surgical work is valued within federal reimbursement structures. The decision by the Center of Medicare and Medicaid Services to place a higher value in prostate cancer management is unknown, however, it is an interesting topic worthy of collective attention. These results raise critical questions about equity in healthcare reimbursement, the methodologies used to assign value to surgical interventions, and the broader implications for medical specialties treating gender-specific malignancies.

The data extracted from CMS reveals a notable contrast, with urologists receiving an average of 59% more work RVUs for prostatectomies compared to breast surgical oncologists for mastectomies per procedure and 12% more per hour of total procedure time (S1 Appendix). Furthermore, the absolute values from CMS showcase a similar pattern, indicating that urologists were allocated an average of 29% more work RVUs for pelvic lymph node removals than breast surgical oncologists for axillary lymph node removals and 3.4% more per hour of total procedure time (S1 Appendix). These findings align with broader systemic trends observed in surgical reimbursement, where procedures predominantly performed on female patients are consistently undervalued.

A recent study examining Canadian healthcare reimbursement structures found that surgeons were compensated 28.1% less for procedures performed on female patients than for comparable procedures on male patients, highlighting structural biases that transcend national healthcare systems [8]. Similar undervaluation trends exist in the United States, as evidenced by studies analyzing gender-based disparities in surgical reimbursement. Research on Medicare payment patterns has shown that female orthopedic surgeons receive significantly lower reimbursements than their male counterparts even after accounting for "time in practice, clinical productivity, practice diversity, subspecialty, beneficiary risk score, and place of service" [9]. Meanwhile, another study found that female neurosurgeons also receive lower Medicare reimbursements despite adjusting for workload and procedural complexity, reinforcing the presence of systemic biases in surgical compensation [10]. These disparities suggest that CMS reimbursement models perpetuate long-standing inequities, influencing not only financial equity but also specialty selection and career sustainability.

Over the last 10–20 years, there are well documented trends in a decrease in work RVUs allocated to surgical oncologists for breast cancer operations. One analysis showed an overall decrease of 15% in medicare reimbursement from 2010–2021 for both breast oncology (-11%) and breast reconstructive surgeries (-16%) respectively [11]. Another study found that from 2003–2023, inflation-adjusted medicare reimbursements for breast cancer-related operations decreased by 20.70% [12]. An overall comparison of medicare reimbursements from 2013–2023 further showed this inflation-adjusted discrepancy in Medicare reimbursements, with a 2.2% decrease for urological procedures versus a 22.4% decrease for breast procedures [13].

Notably, malpractice RVUs for mastectomies were 30% higher than those for prostatectomies (3.81 vs. 2.94), suggesting CMS assigns greater liability risk to breast cancer operations (S1 Appendix). This suggests that breast cancer surgery carries a higher perceived risk in terms of liability. For axillary lymph node removals, however, this higher liability does not translate into greater overall wRVU compensation, further exacerbating the compensation imbalance. If malpractice risk is truly a determining factor in wRVU assignments, then the current reimbursement structure may not be accurately reflecting the financial and legal burdens borne by surgical oncologists treating breast cancer.

Practice expense RVUs (PE RVUs), which account for overhead costs associated with these procedures, were only marginally higher (3.27%) for prostatectomies (11.46) than for mastectomies (11.085) (S1 Appendix). This comparison of PE RVUs was included to control for variabilities in the equipment used for most breast cancer operations vs most prostate cancer operations. The relatively small difference in PE RVUs suggests that differences in resource utilization do not fully explain the wRVU discrepancies observed in our study. Instead, the disparities in wRVUs appear to be driven predominantly by differences in how physician work effort is valued between specialties.

While we have tried to control for the variations in equipment needed by comparing the relative PE RVUs, and control for the variations in expected complications by comparing the malpractice RVUs allocated, these financial comparisons should be viewed within their respective framework. For example, prostate cancer operations continue to rely more heavily on robot assisted surgical services like the Da Vinci robot which requires unique training as well as a higher upfront cost to the practice than what is documented on a per procedure basis [14]. As we continue to further explore these implications, there is growing evidence that these factors are not heavily impactful on CMS' RVU model. The current work RVU allocation by medicare/medicaid focuses more on physician total work times than the actual instrument complexities and physician workload of each subtype of prostatectomy [15]. Thus, the work RVUs as a unit of total operative time is the most accurate independent data points to analyze the importance CMS places on these procedures.

As we continue to ask the question of the value placed on breast cancer vs prostate cancer operations, an in-depth analysis broken down by equipment and training required of only robot assisted surgeries could be beneficial to even further control for these considerations. It is important to also take into consideration the unique training required within the fields of urology and breast cancer operations, and their relative complexities.

### 4.1 Implications

These findings have important implications for healthcare policy and surgical workforce dynamics. If reimbursement disparities persist, they may inadvertently disincentivize physicians from pursuing careers in surgical oncology for breast cancer relative to urology. Between 2000 and 2015, it was found that general surgery residents had a 17% lower exposure to breast cases despite having an overall increase in operative cases [16]. Undervaluation may further discourage attending surgeons from prioritizing these cases and as a result not expose the residents to the cases. Increasing wRVUs for breast procedures could encourage more attending participation, resulting in better resident training and potentially increasing the number of surgeons equipped to manage breast cancer. Given that breast cancer remains one of the most common and impactful malignancies worldwide, ensuring equitable compensation for surgical oncologists is critical for maintaining access to high-quality care for patients. The systemic undervaluation of breast cancer procedures aligns with broader patterns of reimbursement disparities affecting women's health services, which have been documented across multiple surgical specialties [17].

### 4.2 Future direction

Future research should further investigate the rationale behind these wRVU discrepancies, with a focus on surgical complexity, patient outcomes, and economic considerations. Additionally, policymakers should reevaluate the CMS valuation process to ensure fair and accurate compensation for surgical procedures across specialties. For instance, working with physicians and surgeons themselves during the RVU review process can help to ensure evidence-based RVU allocation [18]. Reforming the wRVU model to promote equitable physician reimbursement could lead to a more balanced healthcare system that appropriately values all types of cancer surgery.

## 5. Conclusion

Our study provides evidence of systemic disparities in physician compensation between breast and prostate cancer surgeries. While multiple factors may contribute to these differences, the persistence of these imbalances raises important questions about how surgical procedures are valued and whether changes are needed to promote equity in physician reimbursement. Addressing these disparities is essential for ensuring equitable healthcare delivery and supporting a diverse and motivated surgical workforce.

## Supporting information

**S1 Appendix. CMS physician fee schedule work RVUs, practice expense RVUs, malpractice RVUs and median operative times.**
(XLSX)

## Acknowledgments

The data used in this article have been included and made available in compliance with the principles incorporated by PLOS journal. Supplemental materials obtained from the CMS database have been included for public analysis and can be accessed via the CMS website. All data analysis has been made available in S1 Appendix. I, Noah Khosrowzadeh, had full access to all the data in the study and took responsibility for the integrity of the data and the accuracy of the data analysis. Noah Khosrowzadeh, was the lead author of this manuscript. Aren Saini, Kayla Samimi, Kyle Chambers, Matthew Gompels, Dr. Takita, Dr. Meshman, and Dr. Washington all assisted in supervision and editing of this document.

## Author contributions

**Conceptualization:** Noah Khosrowzadeh, Matthew Gompels, Cristiane Takita, Jessica Meshman, Cyrus Washington.

**Data curation:** Noah Khosrowzadeh, Kyle Chambers, Matthew Gompels, Jessica Meshman, Cyrus Washington.

**Formal analysis:** Noah Khosrowzadeh, Aren Singh Saini, Kayla Samimi, Kyle Chambers, Matthew Gompels, Cristiane Takita, Jessica Meshman, Cyrus Washington.

**Funding acquisition:** Noah Khosrowzadeh.

**Investigation:** Noah Khosrowzadeh, Aren Singh Saini, Kayla Samimi, Kyle Chambers, Matthew Gompels, Cristiane Takita, Jessica Meshman, Cyrus Washington.

**Methodology:** Noah Khosrowzadeh, Kyle Chambers, Matthew Gompels, Cristiane Takita, Jessica Meshman, Cyrus Washington.

**Project administration:** Noah Khosrowzadeh, Aren Singh Saini, Kayla Samimi, Cristiane Takita, Jessica Meshman, Cyrus Washington.

**Resources:** Noah Khosrowzadeh, Aren Singh Saini, Kayla Samimi, Kyle Chambers, Jessica Meshman, Cyrus Washington.

**Software:** Noah Khosrowzadeh, Jessica Meshman, Cyrus Washington.

**Supervision:** Noah Khosrowzadeh, Aren Singh Saini, Kayla Samimi, Kyle Chambers, Matthew Gompels, Cristiane Takita, Jessica Meshman, Cyrus Washington.

**Validation:** Noah Khosrowzadeh, Aren Singh Saini, Kayla Samimi, Kyle Chambers, Cristiane Takita, Jessica Meshman, Cyrus Washington.

**Visualization:** Noah Khosrowzadeh, Aren Singh Saini, Kayla Samimi, Cristiane Takita, Jessica Meshman, Cyrus Washington.

**Writing – original draft:** Noah Khosrowzadeh, Aren Singh Saini, Kayla Samimi, Kyle Chambers, Matthew Gompels, Cristiane Takita, Jessica Meshman, Cyrus Washington.

**Writing – review & editing:** Noah Khosrowzadeh, Aren Singh Saini, Kayla Samimi, Kyle Chambers, Matthew Gompels, Cristiane Takita, Jessica Meshman, Cyrus Washington.

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
