## [Decision Letter · Decision Letter 0]

7 Apr 2025

PONE-D-25-11984Disparities in Physician Compensation for Breast and Prostate Cancer OperationsPLOS ONE

Dear Dr. Saini,

Thank you for submitting your manuscript to PLOS ONE. After careful consideration, we feel that it has merit but does not fully meet PLOS ONE’s publication criteria as it currently stands. Therefore, we invite you to submit a revised version of the manuscript that addresses the points raised during the review process.

**You must address the following comments from Reviewer 2.**1. They are two different types of surgeries. The surgical procedures and the instruments are different. Please discuss more about how to avoid these differences in this research.

2. Are there any historical RVU allocation trends (e.g., CMS updates since 2015) that might influence the conclusion?

3. How might the observed RVU discrepancies influence surgical workforce incentives (e.g., breast cancer specialist shortages) or patient access to care, and what policy reforms could address these issues?============================== Please submit your revised manuscript by May 22 2025 11:59PM. If you will need more time than this to complete your revisions, please reply to this message or contact the journal office at plosone@plos.org . Please include the following items when submitting your revised manuscript:

We look forward to receiving your revised manuscript.

Kind regards,

Lon Jeffrey Van Winkle, Ph.D.

Academic Editor

PLOS ONE

Journal Requirements:

Reviewers' comments:

Reviewer's Responses to Questions

**Comments to the Author**

1. Is the manuscript technically sound, and do the data support the conclusions?

Reviewer #1: Yes

Reviewer #2: Yes

2. Has the statistical analysis been performed appropriately and rigorously? 

Reviewer #1: Yes

Reviewer #2: N/A

3. Have the authors made all data underlying the findings in their manuscript fully available?

Reviewer #1: Yes

Reviewer #2: Yes

4. Is the manuscript presented in an intelligible fashion and written in standard English?

Reviewer #1: Yes

Reviewer #2: Yes

5. Review Comments to the Author

Reviewer #1: The goal of this study was to determine RVU pay discrepancies between surgical oncologists that performed mastectomies vs prostatectomies. The authors found that surgeons performing mastectomies were paid 2.74 RVUs per hour, vs surgeons performing prostatectomies who were paid 3.06 RVUs per hour. Further differences were found between different treatment modalities, which skewed favourably towards surgical oncologists treating prostate cancer

I found this paper to be extremely well written and captivating to read. My only comment is that I found the figures a little blurry and difficult to read, particularly the axis labels in Figure 1 where the writing is smaller. Sharper figures with clearer writing would be preferable. I'd like to congratulate the authors on their excellent manuscript.

Reviewer #2: In this research, the researchers analyzed relative value units in prostatectomies and mastectomies across different treatment modalities. Our main concerns are as the followings:

1. They are two different types of surgeries. The surgical procedures and the instruments are different. Please discuss more about how to avoid these differences in this research.

2. Are there any historical RVU allocation trends (e.g., CMS updates since 2015) that might influence the conclusion?

3. How might the observed RVU discrepancies influence surgical workforce incentives (e.g., breast cancer specialist shortages) or patient access to care, and what policy reforms could address these issues?

6. PLOS authors have the option to publish the peer review history of their article (what does this mean? ). If published, this will include your full peer review and any attached files.

**Do you want your identity to be public for this peer review?** For information about this choice, including consent withdrawal, please see our Privacy Policy .

Reviewer #1: No

Reviewer #2: No

---

## [Author Response · Author response to Decision Letter 1]

11 Apr 2025

Point-by-Point Response to Reviewer Comments

We thank the reviewers for their insightful feedback and are pleased to address each point in detail below. All suggested revisions have been implemented, and exact quotes from the revised manuscript are provided where applicable. We truly appreciate the time you took for such insightful feedback, and we hope to have adequately addressed any questions you may have regarding our submission.

Reviewer #2

Comment 1:

“They are two different types of surgeries. The surgical procedures and the instruments are different. Please discuss more about how to avoid these differences in this research.”

Response:

We appreciate this important point and have expanded our discussion to directly address it. While we recognize that prostatectomies and mastectomies differ in instrumentation and technique, our analysis focuses on work RVUs per total operative time. We included an article cited below to solidify this as the most accurate assessment of urologic procedure work RVUs as the primary determinant used by CMS to assign value to physician effort.

To further control for variability in equipment and facility costs, we compared practice expense RVUs (PE RVUs), which were found to be only marginally higher for prostatectomies (11.46 vs. 11.085 for mastectomies), indicating that equipment-related cost differences alone do not explain the observed discrepancy in physician compensation. We thank you for bringing this question to our attention, and the team felt it was incredibly insightful and important to elaborate on for readers. We hope our additions to our discussions can help address this thought further.

Exact text in manuscript (Discussion):

“While we have tried to control for the variations in equipment needed by comparing the relative PE RVUs, and control for the variations in expected complications by comparing the malpractice RVUs allocated, these financial comparisons should be viewed within their respective framework. For example, prostate cancer operations continue to rely more heavily on robot-assisted surgical services like the Da Vinci robot which requires unique training as well as a higher upfront cost to the practice than what is documented on a per procedure basis. [14] As we continue to further explore these implications, there is growing evidence that these factors are not heavily impactful on CMS’ RVU model. The current work RVU allocation by Medicare/Medicaid focuses more on physician total work times than the actual instrument complexities and physician workload of each subtype of prostatectomy. [15] Thus, the work RVUs as a unit of total operative time is the most accurate independent data point to analyze the importance CMS places on these procedures.”

This clarification demonstrates our effort to minimize the confounding effect of procedural differences and underscores our rationale for using time-adjusted RVUs as a fair comparative metric.

Comment 2:

“Are there any historical RVU allocation trends (e.g., CMS updates since 2015) that might influence the conclusion?”

Response:

Thank you for this thoughtful question. While a thorough literature search did not yield specific historical data on CMS work RVU allocations for individual procedures over time, we have now identified and cited multiple studies examining Medicare reimbursement trends, which are directly based on RVU valuations. These trends are highly informative, as Medicare payments are calculated using the assigned RVUs multiplied by the CMS conversion factor.

Specifically, our revised Discussion includes national data demonstrating a 22.4% decline in inflation-adjusted Medicare reimbursements for breast procedures between 2013–2023, compared to only a 2.2% decline for urological procedures. This suggests a long-standing and widening disparity that aligns with the current RVU-based payment differences highlighted in our study.

To explain why RVU allocation changes may not be publicly traceable year-by-year, we also included the following clarification:

Exact text added to manuscript (Discussion):

“One analysis showed an overall decrease of 15% in Medicare reimbursement from 2010–2021 for both breast oncology (–11%) and breast reconstructive surgeries (–16%) respectively. [11]… From 2003–2023, inflation-adjusted Medicare reimbursements for breast cancer-related operations decreased by 20.70%. [12]… with a 2.2% decrease for urological procedures versus a 22.4% decrease for breast procedures. [13]”

Comment 3:

“How might the observed RVU discrepancies influence surgical workforce incentives (e.g., breast cancer specialist shortages) or patient access to care, and what policy reforms could address these issues?”

Response:

We appreciate this important question and have revised the Implications and Future Directions sections to more directly address how RVU discrepancies may impact training, workforce development, and access to care.

Our updated text discusses how the undervaluation of breast cancer procedures may disincentivize both residents and attending surgeons. This disincentive may contribute to decreased breast case exposure during surgical training, a trend already documented in national data. We argue that increasing wRVUs for breast surgery may incentivize attending surgeons to prioritize these cases, ultimately improving training quality and expanding the future workforce of breast cancer specialists.

Exact text added to manuscript (Implications):

“If reimbursement disparities persist, they may inadvertently disincentivize physicians from pursuing careers in surgical oncology for breast cancer relative to urology. Between 2000 and 2015, it was found that general surgery residents had a 17% lower exposure to breast cases despite having an overall increase in operative cases. [16] Undervaluation may further discourage attending surgeons from prioritizing these cases and as a result not expose the residents to the cases. Increasing wRVUs for breast procedures could encourage more attending participation, resulting in better resident training and potentially increasing the number of surgeons equipped to manage breast cancer. Given that breast cancer remains one of the most common and impactful malignancies worldwide, ensuring equitable compensation for surgical oncologists is critical for maintaining access to high-quality care for patients.”

We also propose a policy recommendation emphasizing the need for greater physician involvement in RVU reassessment, to ensure that procedural complexity, physician workload, and training needs are adequately valued.

Exact text added to manuscript (Future Directions):

“Future research should further investigate the rationale behind these wRVU discrepancies, with a focus on surgical complexity, patient outcomes, and economic considerations. Additionally, policymakers should reevaluate the CMS valuation process to ensure fair and accurate compensation for surgical procedures across specialties. For instance, working with physicians and surgeons themselves during the RVU review process can help to ensure evidence-based RVU allocation. [18]”

We believe these additions clearly address the reviewer’s concerns by linking reimbursement policies to real-world training exposure and long-term workforce sustainability. Thank you for taking the time to thoroughly review our paper. We really appreciate it.

Warm regards,

Aren Saini (on behalf of all co-authors)

University of Miami Miller School of Medicine

arensaini@med.miami.edu

---

## [Editor Report · Decision Letter 1]

16 Apr 2025

Disparities in Physician Compensation for Breast and Prostate Cancer Operations

PONE-D-25-11984R1

Dear Dr. Aren Singh Saini,

We’re pleased to inform you that your manuscript has been judged scientifically suitable for publication and will be formally accepted for publication once it meets all outstanding technical requirements.

Kind regards,

Lon Jeffrey Van Winkle, Ph.D.

Academic Editor

PLOS ONE
---

## [Editor Report · Acceptance letter]

PONE-D-25-11984R1

PLOS ONE

Dear Dr. Saini,

I'm pleased to inform you that your manuscript has been deemed suitable for publication in PLOS ONE. Congratulations! Your manuscript is now being handed over to our production team.

Kind regards,

on behalf of

Dr. Lon Jeffrey Van Winkle

Academic Editor

PLOS ONE